# Structural Characterization of the Plasticizers’ Role in Polymer Inclusion Membranes Used for Indium (III) Transport Containing IONQUEST^®^ 801 as Carrier

**DOI:** 10.3390/membranes11060401

**Published:** 2021-05-27

**Authors:** Alejandro Mancilla-Rico, Josefina de Gyves, Eduardo Rodríguez de San Miguel

**Affiliations:** Departamento de Química Analítica, Facultad de Química, Universidad Nacional Autónoma de México (UNAM), Ciudad Universitaria, Ciudad de México 04510, Mexico; alejmri_1985@hotmail.com (A.M.-R.); degyves@unam.mx (J.d.G.)

**Keywords:** PIM, plasticizer, characterization, indium transport

## Abstract

Polymer inclusion membranes containing cellulose triacetate as support, Ionquest^®^ 801 ((2–ethylhexyl acid) -mono (2–ethylhexyl) phosphonic ester) as extractant, and 2NPOE (o–nitrophenyl octyl ether) or TBEP (tri (2–butoxyethyl phosphate)) as plasticizers were characterized using several instrumental techniques (Fourier Transform Infrared Spectroscopy (FT–IR), Reflection Infrared Mapping Microscopy (RIMM), Electrochemical Impedance Spectroscopy (EIS), Differential Scanning Calorimetry (DSC)) with the aim of determining physical and chemical parameters (structure, electric resistance, dielectric constant, thickness, components’ distributions, glass transition temperature, stability) that allow a better comprehension of the role that the plasticizer plays in PIMs designed for In(III) transport. In comparison to TBEP, 2NPOE presents less dispersion and affinity in the PIMs, a plasticizer effect at higher content, higher membrane resistance and less permittivity, and a pronounced drop in the glass transition temperature. However, the increase in permittivity with In (III) sorption is more noticeable and, in general, PIMs with 2NPOE present higher permeability values. These facts indicate that In (III) transport is favored in membranes with chemical environment of high polarity and efficiently plasticized. A drawback is the decrease in stability because of the minor affinity among the components in 2NPOE–PIMs.

## 1. Introduction

Polymeric inclusion membranes (PIMs) are formed from a solution containing an extractant, a plasticizing agent and a base polymer such as cellulose triacetate (CTA) or polyvinyl chloride (PVC) to obtain a thin film, flexible and stable [1]. PIMs maintain most of the advantages that liquid membranes (LMs) offer, in addition to presenting greater stability and better versatility to obtain optimal efficiency in separation processes. In the last 30 years, membrane-based processes, and especially PIMs, have attracted considerable attention, which is reflected in an exponential increase in the number of articles published [2]. Although the chemical reactions carried out in solute extraction and recovery processes using PIMs are essentially the same as the respective solvent extraction (SX) systems, the transport phenomenon in PIMs and LMs is more complex and it is strongly influenced by both the physicochemical properties of the carrier and those of the solute, as well as the chemical composition of the membrane, and the feed and recovery phases. Within this context, one the main objective of research carried out in PIMs is to maximize the membrane flux while maintaining the efficiency and selectivity of the respective solvent extraction system.

The principal function of one main component of the PIM, the plasticizer, is to improve the flexibility and workability of the polymer. The mode of action of a plasticizer is to penetrate between the polymer chains and “neutralize” the polar groups of the polymer with their own polar groups or by increasing the distance between the polymer chains and thus reduce the action of intermolecular forces [3,4,5,6,7]. The degree of plasticization of a polymer depends largely on the structure and chemical composition of the polymer and the functional groups. There is a wide range of commercially available plasticizers, but only some of them have been used in PIMs systems. Some examples are 2–nitrophenyloctylether (2NPOE), dioctyl phthalate (DOP), polyoxyethylene n-alkyl ethers (POEs), bis(2–ethylhexyl)adipate (DOA), tris(2–ethylhexyl) phosphate (T2EHP), and 2–nitrophenylpentylether (2NPPE) [1]. The relationships between membrane performance, concentration, and the physicochemical characteristics of the plasticizer are complex, entailing a poor understanding of them. Furthermore, such a relationship becomes more complicated if other factors are considered, among which are good compatibility with the polymeric support, low volatility, low viscosity, high dielectric constant, low cost, and low toxicity [1].

In various studies with PIMs, it has been concluded that the increase in the concentration of the plasticizer responds to an increase in the permeability of the metal ion that is being studied (plasticizing effect) [8,9,10]. However, if the concentration range considered is wide, the profile of the curve obtained shows an optimal plasticizer concentration at which the permeability of the metal ion is maximum; beyond this value the flux decreases [11,12,13]. This effect has been attributed to an increase in viscosity, which is not favorable for transport; however, this reason does not explain the fact that beyond the anti-plasticization interval, the addition of the plasticizer causes a decrease in T_g_ and therefore a less viscous medium [1]. In PIMs of CTA, bis(2,4,4-trimethylpentyl)phosphinic acid (CYANEX 272) as carrier and tris(2–ethylhexyl) phosphate (TEHP) and tris(2–butoxyethyl) phosphate (TBEP) as plasticizers, an anti-plasticization effect was observed in the region of low content of plasticizers, where the flow decreases when increasing the amount of plasticizer. In that region the small amounts of plasticizer are sufficient to mobilize the polymeric chains and thus, obtain a more ordered structure that requires less space, hindering the movement of the plasticizer and the carrier [14]. The viscosity of the plasticizer in a PIM is one of the parameters that influence the transport of a solute. Kozlowski and Walkowiak [15] and alternatively Scindia et al. published a correlation between plasticizer viscosity and Cr (VI) flux in CTA and PVC membranes using tri-n-octyl amine (TOA) as carrier [16]. Sugiura and Kikkawa carried out a similar study but with Zn [17].

It is the aim of the present work to shed light on the role that the plasticizer has in PIMs designed for In(III) transport using the commercial reagent Ionquest^®^ 801 through a structural characterization of the polymeric materials using several instrumental techniques (Fourier Transform Infrared Spectroscopy (FT–IR), Reflection Infrared Mapping Microscopy (RIMM), Electrochemical Impedance Spectroscopy (EIS), Differential Scanning Calorimetry (DSC)), together with physical measurements (thickness) and chemical and operational characteristics (components’ compatibility and stability). This research has been carried out to complement previous works related with In (III) transport in PIMs using CYANEX 272 [14], and D2EHPA (bis(2–ethylhexyl) phosphoric acid) [18] as a comparative study among phosphinic (CYANEX 272), phosphoric (D2EHPA), and phosphonic acid (IONQUEST 801) extractants. Since the influence of the plasticizer in transport is dependent on the degree of plasticity, which is largely dependent on the chemical structure of the plasticizer [19], as well as on its interactions with the other components of the membrane system [14,18], it is expected that the generated information will be helpful in the understanding of the plasticizer’s role in PIM systems. To the best of our knowledge a systematic study of the plasticizer’s influence using different commercial organophosphorus derivatives as carriers in PIMs has not been addressed up to now. The Indium (III) system was selected due to the potential use of this metal in electronic devices, which has increased attention in SX-based methods for its recovery [20]. Solvent extraction of indium from different aqueous solutions (sulfate, nitrate, and chloride) by lonquest^®^ 801 has earlier been examined with excellent results [21]. A PIM containing polyvinyl chloride (PVC), and D2EHPA was previously reported for the recovery of In (III) ions from hydrochloric acid medium as well [22].

## 2. Materials and Methods

### 2.1. Reagents

Cellulose triacetate (CTA, Aldrich, Darmstadt, Germany, average MW 966.8 g/mol) as support, Ionquest^®^ 801 ((2–ethylhexyl acid) -mono (2–ethylhexyl) phosphonic ester, Rhodia Inc., La Défense, France) as extractant, and the plasticizers 2NPOE (o–nitrophenyloctyl ether, purity > 99.0%, ε = 23.1, PM 251.33 g/mol, 99% d = 1.041 h = 12.8 cP, Fluka Analytical, Buchs, Switzerland) and TBEP (tri (2–butoxyethyl phosphate), 94%, ε = 8.7, PM 398.48 g/mol 94% d = 1.006 h = 11–15 cP, Sigma-Aldrich, St. Louis, MO, USA) were employed. The solvent used in membrane preparation was dichloromethane (99.9%, J.T. Baker, Allentown, PA, USA).

A 10 mM solution of In (III) in HCl at pH = 0 was prepared from dilution of a stock solution of In at 1000 mg/L obtained by dissolving In_2_O_3_ RA in concentrated HCl (37%, Sigma-Aldrich) to adjust pH. The feed phase was prepared from a dilution of the 10 mM solution with deionized water (MilliQ, 18 MΩ cm). The recovery phase consisted of a 1 M HCl solution, which was prepared by dilution of concentrated HCl (37%, Sigma-Aldrich, St. Louis, MO, USA) with deionized water (MilliQ, 18 MΩ cm).

### 2.2. Instrumentation

To measure the amount of In (III) in transport experiments, a Perkin Elmer 3100 flame atomic absorption spectrophotometer (Waltham, MA, USA) was used at λ of 303.9 nm, aperture of slit of 0.7 nm, and an air–acetylene gas mixture using an impact flow spoiler device. The pH measurements were carried out with a Corning 440 digital pH meter with a Pinnacle glass electrode (Corning, NY, USA). The agitation of the samples in the liquid–solid extraction experiment was carried out in 50 cm^3^ centrifuge tubes with a Burrell model 75 wrist action mechanical shaker (Burrell Scientific Inc, Pittsburgh, PA, USA). The thickness of the membranes was measured with a Fowler IP54 electronic micrometer (Fowler High Precision, Newton, MA, USA). Impedance measurements were performed with a Solartron (Hampshire, UK) SI 1287 potentiostat coupled to a Solartron SI 1260 computer-controlled impedance analyzer using the Zplot version 2.0 b software from Scribner Associates Inc. (Southern Pines, NC, USA). Spectra and infrared maps were obtained using a Perkin–Elmer GX-FTIR spectrometer coupled to an Autoimage FTIR microscope with Autoimage 5.0 software. The thermal behavior of the membranes was analyzed in a Mettler TC15 differential scanning calorimeter (DSC, Herisau, Switzerland).

### 2.3. Synthesis of Polymeric Inclusion Membranes

The method for preparing the PIMs of CTA–2NPOE/TBEP–Ionquest^®^ 801 consisted of weighing a specific amount of each of the membrane components and dissolving them in 5 mL of dichloromethane under constant stirring in an approximate time of 45 min. Once a homogeneous and translucent mixture was obtained, it was poured into a glass petri dish (6 cm in diameter) to allow the solvent to evaporate at room temperature for 24 h. Finally, a thin and transparent film was obtained that can be detached from the petri dish by adding a small amount of cold water. This method was proposed by Hayashita [23]. A description and schematic representation of the evaporation casting method can be found in the literature [24]. The thickness of the membrane was determined by 5 uniformly distributed measurements.

### 2.4. Transport Experiments

In (III) permeability experiments were carried out in a transport cell consisting of two cubic compartments with a maximum volume of 90 cm^3^ separated by a female–male type connector of circular shape with 2.32 cm of diameter where the membrane was placed, with an exposure area of 3.64 cm^2^. The time of the transport experiment was 3 h under constant stirring at 550 rpm, taking a 1.5 mL aliquot every 30 min from both phases. The side of the membrane exposed to air during synthesis was faced to the feed phase (0.1 mM In (III) in HCl pH = 2.0), while the opposite side was faced to the recovery phase (1M HCl) during the transport experiments. Transport conditions and sampling times were selected according to previously reported similar systems [14,18].

### 2.5. In (III) Quantification

For In (III) quantification, a five-point calibration curve was prepared from a 100 ppm In stock solution in a concentration range from 2 to 10 ppm. This stock solution was prepared from a standard solution for atomic absorption of In with a concentration of 1005 μg/mL in 1% by weight of HNO_3_ (Aldrich).

### 2.6. Liquid–Solid Extraction Experiments

The liquid–solid extraction consists of the distribution of the analyte between a solution of 0.1 mM In (III) with pH between 1.8 and 2.0 (feed phase) and a PIM of CTA–2NPOE/TBEP–Ionquest^®^ 801. For the experiment, 10 mL of the In (III) feed phase were placed together with the membrane in a 50 mL centrifuge tube at a moderate shaking level for 2 h and, subsequently, a 1.0 mL aliquot of the liquid phase was withdrawn for analysis.

### 2.7. Membrane Stability

The stability of the PIMs was evaluated by testing several membranes with varying plasticizer and extractant compositions, setting the amount of CTA at 30 mg.

### 2.8. EIS Characterization

Measurements were performed on a cell adapted to the equipment in the AC impedance mode with two 25 cm^2^ stainless steel electrodes and an exposed area of 1 cm^2^; the frequency sweep was 0.1 Hz to 1 MHz. The applied voltage was 10 mV of alternating current amplitude. The measurements were carried out by EIS to the PIMs equilibrated with 40 mL of the feed phase for 5 and 180 min. Not equilibrated PIMs were measured as well.

### 2.9. RIMM Characterization

Membrane mapping was carried out with an aperture of 100 µm × 100 µm in an area of 1000 µm × 1000 µm, with a resolution of 6 cm^−1^ and 30 scans per point in the region of 4000 to 700 cm^−1^. The 2NPOE distribution was determined using the 1528 cm^−1^ band, while for TBEP and Ionquest^®^ 801 the 1136 cm^−1^ and 801.976 cm^−1^ bands were used, respectively.

### 2.10. DSC Characterization

The amount of sample analyzed was 10 mg. The experiments were carried out at a heating rate of 10 °C/min from 25 to 300 °C under nitrogen atmosphere. The temperature program was established based on a similar system [18].

## 3. Results and Discussions

### 3.1. Evaluation of PIM Formation

PIMs are systems that are composed of a polymeric support, a plasticizer, and an extracting agent, and as such, those elements can be visualized in a ternary diagram. In this way, the formation of the polymeric membrane was evaluated by testing multiple CTA, plasticizer (2NPOE or TBEP), and Ionquest^®^ 801 compositions. 30 mg of CTA was taken as reference, as good mechanical properties and formation characteristics had been previously observed using such amounts [14,18]. The results are shown in Figure 1A,B. It was considered that an adequate PIM formation gives rise to a uniform polymeric film, free of visible pores, transparent and colorless (black squares). In the same figure, photographs of some samples are shown. As can be seen in the ternary diagram, there is an area where PIMs are not well–formed (red dots). This area includes CTA compositions approximately lower than 30% *w*/*w*, greater than 45% *w*/*w* of Ionquest^®^ 801, and lesser than 50% *w*/*w* of 2NPOE and 30% *w*/*w* of TBEP (upper plots). However, before making any comparison, due to the difference in molar weight of the plasticizers, the diagrams were redrawn in terms of mole percent (lower plots). The areas for no favorable PIM formation in these coordinates then were: CTA < 15%, Ionquest^®^ 801 > 40% and 2NPOE < 50% and CTA < 15%, Ionquest^®^ 801 > 55% and TBEP < 35%. These results clearly indicated that PIMs with TBEP accepted more Ionquest^®^ 801 and required less plasticizer content in comparison to those with 2NPOE for a positive formation, indicating a better affinity of Ionquest^®^ 801 for TBEP than for 2NPOE. This makes sense considering the similarity of the structure between the organophosphorus compounds.

### 3.2. PIM Thickness Characterization

One of the physical factors that significantly influence the transport of particles through membranes is thickness, as the driving force, to which the transport of a particle is due, is determined by the quotient between the concentration gradient and the thickness of the membrane according to Fick’s law. In Figure 2A,B PIMs’ thicknesses, corresponding to the average obtained from 5 measurements at different points on the membrane, are shown. As for PIMs with CTA–2NPOE–Ionquest^®^ 801, thicker membranes had compositions with CTA < 10%, Ionquest^®^ 801 < 35%, and 2NPOE > 50% mole/mole and CTA < 10%, Ionquest^®^ 801 < 60%, and TBEP > 40% mole/mole. These results indicated that thicker PIMs are obtained with less plasticizer and Ionquest^®^ 801 contents with TBEP than those with 2NPOE which may be related to the characteristic plasticization properties of the films.

### 3.3. Liquid–Solid Extraction Characterization

Liquid–solid extraction experiments were carried out to obtain information about the stoichiometry of the metal–extractant complex and its extraction equilibrium constant on the membrane [18,25]. The distribution coefficient D between the membrane and the aqueous phase is defined according to Equation (1):(1)$$D=\frac{{\left[\mathrm{In}\left(\mathrm{III}\right)\right]}_{\mathrm{membrane}}}{{\left[\mathrm{In}\left(\mathrm{III}\right)\right]}_{\mathrm{aqueous}}}=\frac{\left(C_{0}-C_{f}\right)V_{\mathrm{aq}}}{{\mathrm{MC}}_{f}}$$
where C_0_ represents the initial concentration of the metal in the aqueous phase, C_f_ is the equilibrium concentration in the aqueous phase, V_aq_ is the volume of the aqueous phase, and M represents the weight of the membrane. In this experiment, two levels of the amount of plasticizer in the membrane were employed, 0.01 g and 0.09 g, while the amount of CTA remained fixed at 30 mg with variable amounts of Ionquest^®^ 801, from 0.0009 to 0.0029 g (Table 1).

Once the value of the distribution coefficient D was experimentally evaluated, the LETAGROP–DISTR program [26] was used to determine the stoichiometry of the metal–extractant complex and the value of the extraction constant K_E_. This program searches for the best equilibrium constants that minimize the error squares sum defined by Equation (2):(2)$$U=\sum_{N_{p}}{\left({\mathrm{logD}}_{\mathrm{calc}}-{\mathrm{logD}}_{\exp}\right)}^{2}$$
where D_exp_ is the distribution coefficient experimentally established and D_calc_ is the value calculated by the program assuming a given chemical model. The program also calculates the standard deviation σ (log D) defined by Equation (3):(3)σ(log D) = (UNp−Nk)12
where N_p_ is the number of experimental points and N_k_, number of equilibrium constants.

The equilibria considered in the calculations are reported in Table 2 and the best models for In (III) extraction are presented in Table 3. According to the proposed extraction models, for both plasticizer and concentration levels, the In (III) extraction mechanism is explained through the formation of the complex InR_3_ between the analyte and the extractant with different solvation degrees by the latter compound. The extraction constants (K_E_) obtained were very similar for both plasticizers; however, K_E_’s values obtained at a low level of plasticizer were higher than those with high level. The species found in the literature in nitrate medium and using PIMs with an organophosphinic carrier homologous to Ionquest^®^ 801 (CYANEX 272) and 2NPOE and TBEP as plasticizers were InR_3_2HR for a 55% *w*/*w* concentration of plasticizer and InR_3_ for 73% *w*/*w* [14]. For a PIM with an organophosphoric carrier homologous to Ionquest^®^ 801 (D2EHPA) and TBEP as plasticizer, InR_3_2HR has been reported for 25% *w*/*w* of plasticizer and InR_3_4HR for 75% *w*/*w*. The presence of these two species at the studied TBEP concentrations was attributed to more favorable solvation of the extracted complex as the plasticizer content increases due to the decrease in the dielectric constant of the medium [18].

### 3.4. In (III) Permeability Profiles

It was previously mentioned that the effect of a plasticizer in the PIM allows improving its mechanical properties and durability. In addition, it has been reported that this parameter impacts its permeability as well [13,14]. To study the effect of the plasticizer in the CTA–2NPOE/TBEP–Ionquest^®^ 801 systems, In (III) permeability was determined through 5 membranes with a fixed amount of 30 mg of CTA and Ionquest^®^ 801, and variable amounts of plasticizer from 10 to 50 mg (Table 4 and Figure 3). The permeability was evaluated according to Equation (4):(4)$$\mathrm{In}\frac{{\left[\mathrm{In}\left(\mathrm{III}\right)\right]}_{\mathrm{feed}}}{{\left[\mathrm{In}\left(\mathrm{III}\right)\right]}_{\mathrm{initial}}}=-\mathrm{Pt}\frac{A}{V}$$
where [In(III)]_feed_ represents the concentration of In (III) in the feed phase at time t, [In (III)]_initial_ represents its initial concentration, P is the permeability, t the time, A represents the exposed membrane area, and V the volume of the transport cell.

The permeability profile as a function of the amount of plasticizer is similar for both systems, showing a parabolic curve shape, where a minimum permeability value is located. Such trends in permeability profiles have also been observed in PIMs with CTA, CYANEX 272 as carrier in the presence of plasticizers such as 2NPOE, TBEP, TEHP, or CTA–TBEP–LIX^®^ membranes for In (III) and Cu (II), respectively, and in CTA–2NPOE–Kelex 100 membranes for Au (III) transport [13,14,28]. These permeability profiles present two zones: one with low plasticizer content, where an anti-plasticization effect is observed (decrease in permeability by increasing the amount of plasticizer, due to the fact that the small amount of plasticizer present is sufficient to immobilize the polymeric chains, generating a more ordered structure that requires less space and consequently does not favor the mobility of the carrier and the plasticizer), and one in which a plasticization effect is noticed (increase in permeability as the amount of plasticizer increases as, in these circumstances, the plasticizer makes the membrane a better medium for the mobility of the extractant). According to the results presented before related to PIM favorable formation, the plasticization zone for TBEP is observed with a lesser amount of plasticizer in comparison to 2NPOE. Membranes in the absence of plasticizer did not present In (III) permeability; however, a minimum absorption of the analyte of about 1 ppm was observed after 180 min of pertraction, but In was retained within the membrane phase.

### 3.5. FT-IR Characterization

To evaluate the interactions between the different components of the PIM, the characterization of the membranes was carried out by means of infrared spectroscopy (Figure 4). The most important distinctive signals are listed in Table 5; these signals were also reported in the literature for similar PIMs [14,29,30,31]. The compositions of the studied PIMs are reported in Table 6.

The IR spectra for the m2 and m4 PIMs were practically the same because they had the same components (CTA, 2NPOE, Ionquest^®^ 801) with the only difference that the bands that appear in the region of 1400 to 1600 cm^−1^ were more intense for the m4 sample, most likely because it contained a greater amount of 2NPOE. The IR spectra for both m2 and m4 PIMs showed characteristic bands for each of the components present. For the spectra of the membranes with CTA–2NPOE–Ionquest^®^ 801, bands at 2958, 2930, 2859, and 2873 cm^−1^ were observed and assigned to the characteristic vibrations of CTA C–H bonds or also to –CH2– of the aliphatic chain of 2NPOE; the band at 1755 cm^−1^ is characteristic of a carbonyl group C=O of CTA, the signal at 1528 cm^−1^ is characteristic of the NO_2_ group of the plasticizer, while the band at 1367 cm^−1^ confirmed the vibrations of C-H bonds in CTA. The band at 1235 cm^−1^ is another characteristic band of 2NPOE associated with the vibration of the R–O–CH_2_ bond, and the bands associated with Ionquest^®^ 801 are 1047 cm^−1^ with an attached signal at 985 cm^−1^ due to the vibration of the P–OH bond present in the extractant.

Similar to the PIM of CTA–2NPOE–Ionquest^®^ 801, the doublet-shaped band at 2959, 2933, and the one present at 2873 cm^−1^ were associated with vibrations of C–H bonds of CTA. The band at 1756 cm^−1^ was associated with a C=0 carbonyl group of CTA while the signal present at 1368 cm^−1^ could be a confirmation band of the vibrations of the C–H bonds present in the CTA. The signal at 1043 cm^−1^ could be associated with the vibration of the C–O–C bond in CTA or the vibration of the P–O–C bond present in Ionquest^®^ 801 or in TBEP. The band that appears at 987 cm^−1^ was also associated with a vibration of the P–OH bond in the extractant.

The band at 1459 cm^−1^ is a characteristic band for TBEP, however, it is not clear what type of vibration it corresponds to. It is also important to note that the band at 1235 cm^−1^, previously assigned to 2NPOE, in the IR spectra for the membranes with CTA–TBEP–Ionquest also appear, such that the assignment of this band is somewhat uncertain.

### 3.6. RIMM Characterization

To know the distribution profiles of the components (plasticizer and extractant) within the membrane, Infrared Reflection Mapping Microscopy was used. The composition of the membranes analyzed by this technique is reported in Table 6. To isolate the distribution profiles of each component of the membrane, the mapping was carried out with membranes that included only the polymeric support (CTA) and one of the two components, either the extractant or the plasticizer, present in equal amounts. The distribution profiles were carried out considering both faces of the membrane distinguishing between the upward face (face 1) and the downward face (face 2) assigned according to the synthesis procedure.

It is important to mention that to compare the membranes among each other, a normalized absorbance scale was established in the distribution profiles, thus taking the minimum and the maximum values obtained from the absorbance in the maps as limits of the scale. The 2NPOE distribution profile in membrane 41 is presented in Figure 5A (1 and 2). In general terms, 2NPOE distribution on both faces is very similar and is practically uniformly distributed along the mapped area. The distribution profile of Ionquest^®^ 801 in membrane 39 is presented in Figure 5B (1 and 2). In this profile, a slightly greater amount of Ionquest^®^ 801 can be observed on face 1. With respect to the 2NPOE profile, the presence of Ionquest^®^ 801 on both sides of the membrane is lower, and more irregular distributions are observed. The TBEP distribution profile in membrane 40 is presented in Figure 5C (1 and 2). The TBEP distribution profile in the CTA is to some extent different on both sides of the membrane; a slightly lower quantity and an irregular distribution can be seen on side 2, while side 1 had a more homogeneous profile. Interestingly, Ionquest^®^ 801 and TBEP, both organophosphorus compounds, had similar distribution profiles.

The 2NPOE distribution profile in the m2 membrane is shown in Figure 6A (1 and 2). Comparing the distribution profile of 2NPOE in membrane 41 and the profile obtained in membrane m2, it can be observed that there was a strong change in the distribution of 2NPOE; while in the first profile a fairly regular and appreciable distribution was observed; in the second one, the presence of 2NPOE on face 1 decreased appreciably, presenting an irregular dispersion. On face 2 a greater presence of 2NPOE is observed without having a regular distribution. A lower presence of 2-NPOE was expected in this case, as less plasticizer was used in m2 than in membrane 41. The distribution profile for Ionquest^®^ 801 in the m2 membrane is shown in Figure 6B (1 and 2). Comparing the distribution profiles of Ionquest^®^ 801 in membrane 39 and in membrane m2, notable changes were observed, even though an equal amount of extractant was used in both membranes. In membrane m2, the presence of Ionquest^®^ 801 on both faces of the membrane was lower than in membrane 39. This difference was more accentuated on face 1. In addition, the distribution of the extractant was much more irregular in membrane m2, where areas of absence of the extractant and areas of small conglomerates were even observed. The distribution profile between both components, Ionquest^®^ 801 and 2NPOE, was very similar; a greater presence of plasticizer than extractant can be observed on both sides of the membrane. However, it can be said that the presence of 2NPOE in the m2 membrane affected the distribution profile of Ionquest^®^ 801, obtaining a more irregular profile and a lower presence of the extractant on the faces of the membrane, even though this component is in a higher proportion than the plasticizer. The 2NPOE distribution profile of the m4 membrane components is shown in Figure 6C (1 and 2). The use of a greater amount of plasticizer in the m4 membrane is reflected in the distribution profile of 2NPOE, as both sides of the membrane show a fairly regular and homogeneous distribution of this component, and even clumps of the plasticizer were seen on face 2. The Ionquest^®^ 801 distribution profile of the same PIM is shown in Figure 6D (1 and 2). Even though the same amount of Ionquest^®^ 801 was used in both membranes (m2 and m4), the distribution profile of Ionquest^®^ 801 was more uniform and regular in m4 than in m2, and an even greater amount of the extractant was appreciated. Again, the presence of 2NPOE in the membrane modified the distribution profile of the extractant. On this occasion, as the plasticizer was in a higher quantity, the best dispersion of the extractant on both sides of the membrane was favored, probably due to a dragging effect, without both components necessarily being compatible and soluble with each other.

For m7 and m9 PIMs, it was not possible to find a band that allowed obtaining the independent distribution profile of Ionquest^®^ 801, given the overlap with the CTA or TBEP bands. The m7 distribution profiles of TBEP are shown in Figure 7A (1 and 2). According to the obtained profiles, a higher concentration was observed on side 1, while a more homogeneous distribution of TBEP was observed on side 2. Comparing this profile with the distribution of TBEP on membrane 40, a notable difference was appreciated in the amount of this plasticizer present on side 1, which is lower, despite the fact that the PIM had more TBEP than m7. The distribution profiles of TBEP in the m9 membrane is shown in Figure 7B (1 and 2). As observed, the distribution profiles on both sides of the membrane was quite homogeneous. A higher concentration of TBEP was more present in the m9 membrane than in the m7 and was reflected in a greater presence of this plasticizer on both sides of the membrane, especially on side 2, as well as in a more homogeneous distribution of this component.

In conclusion, in the presence of CTA, 2NPOE was more homogeneously distributed than TBEP, while TBEP and Ionquest 801^®^ showed analogous profiles due to their similarity in structure. However, when ternary systems were formed, distributions changed. TBEP showed more affinity for Ionquest 801^®^ than 2NPOE and the compatibility of the extractant and plasticizer increased with the augment in plasticizer content for both systems. These results are consistent with those reported before related to favorable PIM formation experiments (Section 3.1).

### 3.7. EIS Characterization

The electrochemical impedance spectroscopy (EIS) technique is an electrochemical method that has been applied for PIM characterization [14,32] based on the use of an alternating current (AC) signal that is applied to a system and the corresponding response at different frequencies is determined. Four membranes were used for which their compositions are indicated in Table 7.

The equivalent circuit used to model the data obtained for the PIMs is shown in Figure 8.

This circuit is based on two equivalent circuits used to model a PIM system like the one studied [14,18]. The Nyquist plots obtained for the membranes are presented in Figure 9.

The adjustment of the data obtained through the equivalent circuit is presented in Table 8.

The dielectric constant or relative electrical permittivity of the membrane, ε_r,m__,_ was calculated according to the literature [32], using Equations (5) and (6):(5)$$C=\frac{1}{2{\pi \mathrm{fR}}_{\mathrm{memb}}}$$
(6)$$\varepsilon_{r}=\frac{\mathrm{dC}}{{A\varepsilon }_{0}}$$
where C represents the capacitance (Farads), f is the frequency (the maximum value in Hertz of the Nyquist graph), R_memb_ is the resistance of the membrane (in ohms), ε_0_ is the relative electrical permittivity of empty space (8.854 × 10^12^ F/m), A represents the exposed membrane area in m^2^, and d the membrane thickness in m. From the Nyquist graphs it was seen that none of the spectra of the membranes reached a semicircle; in general, the spectra showed a partial semicircle, a situation that occurs when the RC ratio is large. Inspecting the data in relation to the composition of the membranes and the values for R_memb_ and ε_r,m_, in the case of CTA–2NPOE–Ionquest^®^ 801 membranes, the concentration of the plasticizer influenced the increase much more in R_memb_ augment and in the decrease of ε_r,m_ than that of the extractant, while in the case of membranes with TBEP the increase in the concentration of Ionquest^®^ 801 influenced more in the increase of R_memb_. In the case of ε_r,m_, the values were very constant, i.e., it seems that there was no appreciable influence of the concentrations of plasticizer and extractant.

In general, the PIMs of CTA–2NPOE–Ionquest^®^ 801 presented higher resistances than those of CTA–TBEP–Ionquest^®^ 801, a fact that is confirmed in the literature for a similar study with the homologous organophosphoric carrier [14].

Once the membranes were equilibrated with In (III) at different times (5 and 180 min), the general trend towards a decrease in R_memb_ could be seen. This inversely proportional relationship between permeability and resistance has already been observed in reported studies [14]. The dielectric constant of the studied membranes, ε_r,m_, in general presented higher values for the CTA–TBEP–Ionquest^®^ 801 membranes than for those for 2NPOE, which agreed with that reported in the literature [14]. The ε_r,m_ also presented variations with the adsorption of In(III); as for the membranes with 2NPOE, the ε_r,m_ increased as the adsorption of In (III) increased, while in the case of membranes with TBEP this trend was also observed; however, it was not as significant as in the case of membranes with NPOE. The previous trend can be explained by considering the values of the dielectric constants of the plasticizers ε_r,NPOE_ = 23.1 and ε_r,TBEP_ = 8.7, in which, the value of ε_r,m_ for 2NPOE favored in a better way a chemical environment of higher polarity than in the case of membranes with TBEP.

### 3.8. Thermal Analysis Characterization

The permeability, mechanical, chemical, and thermal properties of the PIMs are highly related to the polymeric state of the membrane. The glass transition temperature, T_g_, is very important data in polymers, ceramics, and glasses, as it describes the change of state of a polymer. Below such temperatures, the polymer takes on a rigid glassy structure; above it the polymer acquires an amorphous structure. In the case of a PIM, the rigid state of its polymeric backbone limits the transport properties of the membrane. Plasticizers are very effective agents that lower the glass transition temperature of the polymer. The degree to which the T_g_ is reduced depends on the thermodynamic compatibility of the plasticizer and the polymer. The higher this compatibility, the more plasticizer can be added to the polymer formulation, which also increases lubricity. The efficiency in plasticization and the thermodynamic compatibility of plasticizers and polymers depend on properties such as chemical structure, molecular weight, functional groups, size of aliphatic chains, and diffusion and solubility of plasticizers. T_g_ is an indirectly descriptive parameter for the transport properties in a PIM. Low T_g_ polymers have higher mobility.

The study carried out by Rodríguez de San Miguel et al. [18] showed that T_g_ is a parameter closely related to the structure of the polymer and as such related to the composition of the PIM. In that study, a clear trend of the decrease in T_g_ was observed when increasing the amount of TBEP. In addition, a correlation with a more efficient indium transport for membranes having higher concentrations of plasticizer, as the more plastic the membrane become (lower T_g_), the higher the permeability [18]. The compositions of PIMs and T_g_ determinations for CTA–2NPOE/TBEP–Ionquest^®^ 801 membranes are presented in Table 9.

According to the values, the effect of going from one region to another (anti-plasticization → plasticization) was very pronounced in the case of NPOE, while in the case of TBEP it was less perceptible. This observation is consistent with the changes in permeability observed in the comparison graph of both plasticizers (Table 9, last column). This passage from one region to another was related to the number of mole of the plasticizer present in the membrane; in the case of membranes with TBEP, the concentration of the plasticizer was approximately half that of 2NPOE in its respective system. As studied before, the amount of plasticizer present is highly related to the T_g_ and the permeability of the membrane. An increase in the amount of plasticizer within the membrane causes a decrease in T_g_ and a higher permeability. Considering the permeability values of membranes 2, 4, and the T_g_ of membranes 35 and 36 with 2NPOE (Table 9), it was found that T_g_ decreased appreciably while the permeability increased significantly with increasing the concentration of 2NPOE. This can be attributed to structural factors of the membrane, where a more plastic environment favors the mobility of the complex [29].

In PIMs with TBEP, despite increasing the concentration of the plasticizer from 0.59 to 0.96 mmol/g, the permeability of membranes 7 and 9, and the T_g_ of membranes 37 and 38 (average values), did not appreciably change. This means that there was no substantial structural change in the considered membranes even though there was a higher concentration of TBEP in one of them. This factor may be attributed to the similarity in structures of TBEP and Ionquest^®^ 801, and a possible change in the transport mechanism with plasticizer concentration that lead to a slightly noticeable transition but with similar permeation. With 2NPOE the presence of the polar groups in its chemical structure provided a substantial change in the chemical environment of the PIM. An inverse relationship between plasticization efficiency and molecular weight of the plasticizer has been reported in some systems, Ref. [19] being the case observed.

### 3.9. PIM Stability Characterization

One of the main advantages that PIMs offer compared to other types of membranes is their stability during operation for several cycles, i.e., their lifetime. In the present case, the number of operating cycles was established for 2 different membranes of each PIM system. Membranes 2 and 4 were used as representative ones with 2NPOE, while membranes 7 and 9 for membranes with TBEP. To determine the operating cycles, a continuous repetition of the transport experiment was carried out on the same membrane, renewing the feed and recovery phases after each cycle. The results are shown graphically in Figure 10. In the graphs of the operating cycles for membranes 2, 4, 7, and 9, PIMs 7 and 9 are the only ones that maintained a constant permeability value during 10 cycles. In membrane 9 permeabilities between 0.04 and 0.05 cm/min were observed, while membrane 7 had permeabilities between 0.02 and 0.04 cm/min. On the contrary, PIMs 2 and 4 presented maximum permeability values during the first cycle, and then a drop in permeability was observed, after which it remained constant (0.04 ± 0.01 cm/min in the case of membrane 2 and 0.07 ± 0.01 cm/min in the case of membrane 4). These results may be explained considering a better interaction between Ionquest^®^ 801 and TBEP than that with 2NPOE which avoided possible exudation of the PIM components.

## 4. Conclusions

Composition regions were established for the synthesis of the membranes in a wide range. The areas for non-favorable PIM formation were for 2NPOE and TBEP, respectively: CTA < 15%, Ionquest^®^ 801 > 40% and 2NPOE < 50%, and CTA < 15%, Ionquest^®^ 801 > 55% and TBEP < 35% mole/mole. PIMs with TBEP accepted more Ionquest^®^ 801 and required less plasticizer content in comparison to those with 2NPOE for a positive formation, indicating a better affinity of Ionquest^®^ 801 for TBEP than for 2NPOE. PIM thickness measurements supported this statement. In FT–IR characterization, in general, signals of the functional groups of the membrane components were found, and the formation of covalent bonds was not appreciated. RIMM analyses showed that the distribution of 2NPOE on the polymeric support was homogeneous; for TBEP and Ionquest^®^801 irregular distributions were obtained, and a lower presence of these organophosphate components with respect to those with 2NPOE. However, when ternary systems were formed, distributions changed. TBEP showed more affinity for Ionquest 801^®^ than 2NPOE and the compatibility of the extractant and plasticizer increased with the augment in plasticizer content for both systems. Liquid–solid extraction experiments allowed the identification of the extraction reactions and equilibrium constant values, which did not appreciably change with the type of plasticizer used. Permeability and resistance values, characterized with EIS, are inversely proportional parameters. As for the 2NPOE membranes, the increase in the concentration of the plasticizer influenced the increase in R_mem_ and the decrease in ε_r,m_ much more than the extractant. As for TBEP–PIMs, the increase in the concentration of Ionquest^®^801 (from 0.33 mmol/g to 1.08 mmol/g) had a greater influence on the increase in R_mem_, while the ε_r,m_ values were constant. The dielectric constant of the studied membranes, in general, presented higher values for TBEP–PIMs than for those for 2NPOE. Initially, the 2NPOE–PIMs presented higher average resistance than those of TBEP; however, as the contact time with the cation solutions increased, they tended to decline for both plasticizers, this effect being higher for TBEP than for 2NPOE, which is congruent with the differences in permeability (2NPOE > TBEP). The ε_r,m_ also presented variations with the adsorption of In (III); it increased as the adsorption of In (III) increased for both plasticizers. However, this trend was not as pronounced for TBEP as it was for 2NPOE, which can be justified by their difference in value of ε_r,m_, which favored in a better way a chemical environment with higher polarity for 2NPOE. The better stability of TBEP PIMs during 10 continuous cycles of operation was assigned to a better chemical interaction between TBEP and Ionquest^®^ 801, which would explain why the extracting agent is better retained within the membrane.

In comparison to TBEP, 2NPOE presented then less dispersion and affinity in the PIMs, a plasticizer effect at higher content, higher R_mem_ and less ε_r,m_, and a pronounced drop in the T_g_ values. However, as In(III) was absorbed by the PIM, these parameters changed and an increase in ε_r,m_ and a decrease in R_mem_ were observed, this effect being more pronounced for 2NPOE than for TBEP. In conjunction all the information suggested a better plasticization efficiency of NPOE, which seems to be phase separated, that in the presence of the cation gave rise to a medium of high mobility and polarity, where the structural change promoted by the plasticizer is a key factor in the transport efficiency of the PIM system. A drawback was the decrease in stability because of the minor affinity among the components in 2NPOE–PIMs.

## Figures and Tables

**Figure 1 membranes-11-00401-f001:**
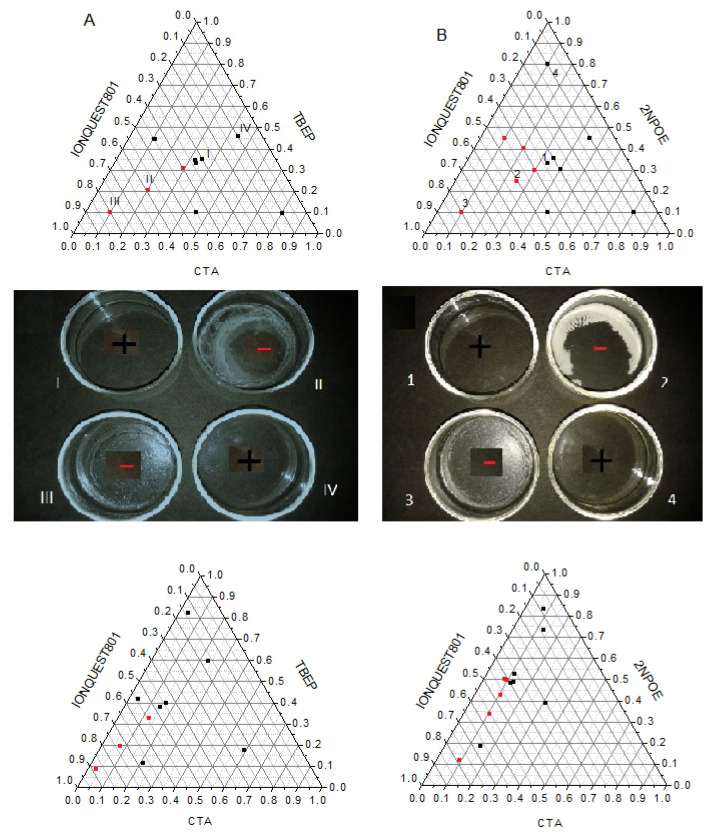
Ternary diagrams for PIMs formation using 2NPOE (**A**) and TBEP (**B**) as plasticizers. Black squares stand for a positive (+) membrane formation while red dots for a negative (−) one. Upper diagrams correspond to weight percent while lower diagrams to mole percent.

**Figure 2 membranes-11-00401-f002:**
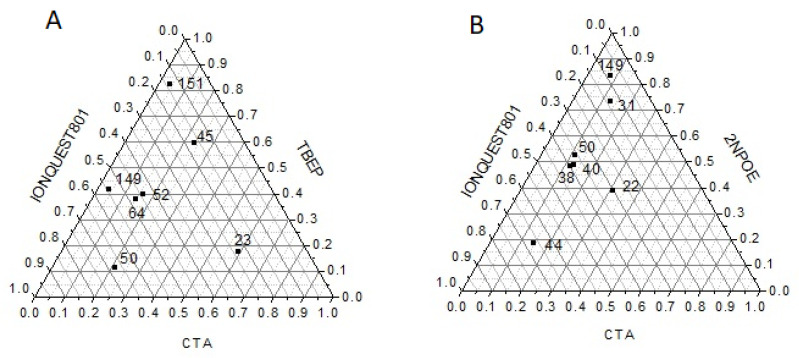
Ternary diagrams (mole percent) for PIMs with TBEP (**A**) and 2NPOE (**B**) denoting the magnitude in their thicknesses (μm, numbers inside) as a function of their composition.

**Figure 3 membranes-11-00401-f003:**
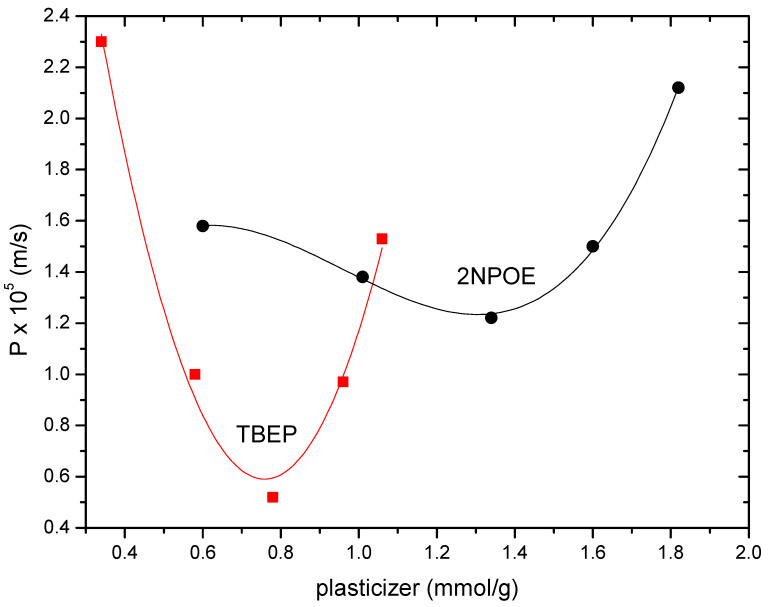
Indium (III) permeability profiles for PIMs 1–5 (2NPOE) and 6–10 (TBEP) at variable plasticizers’ concentration.

**Figure 4 membranes-11-00401-f004:**
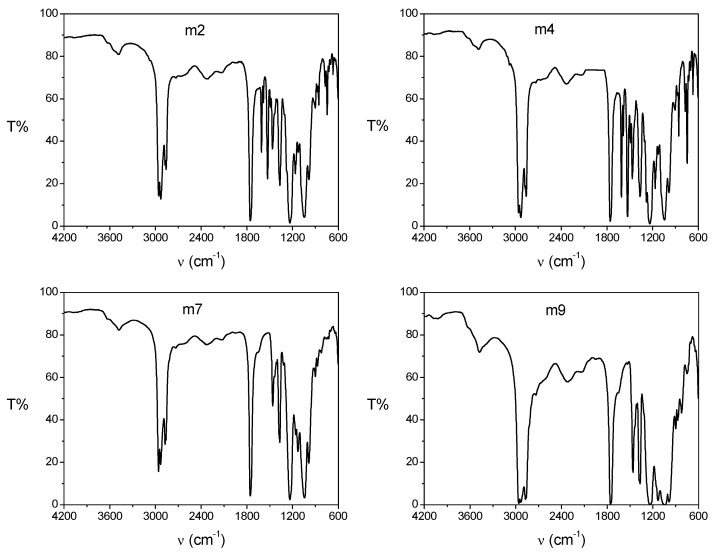
FTIR spectra of PIMs m2, m4, m7, and m9 with compositions indicated in Table 6.

**Figure 5 membranes-11-00401-f005:**
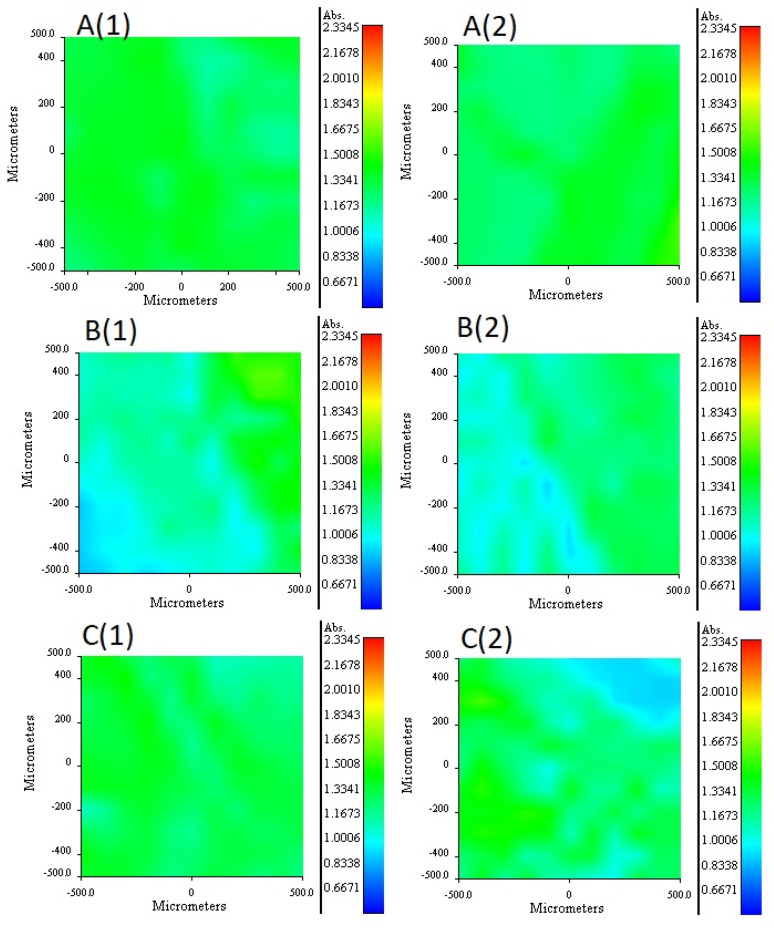
RIMM maps monitoring the distribution of 2NPOE in PIM 41 (**A**), Ionquest^®^ 801 in PIM 39 (**B**), and TBEP in PIM 40 (**C**). (1) stands for face 1 and (2) for face 2 of the PIMs (see text).

**Figure 6 membranes-11-00401-f006:**
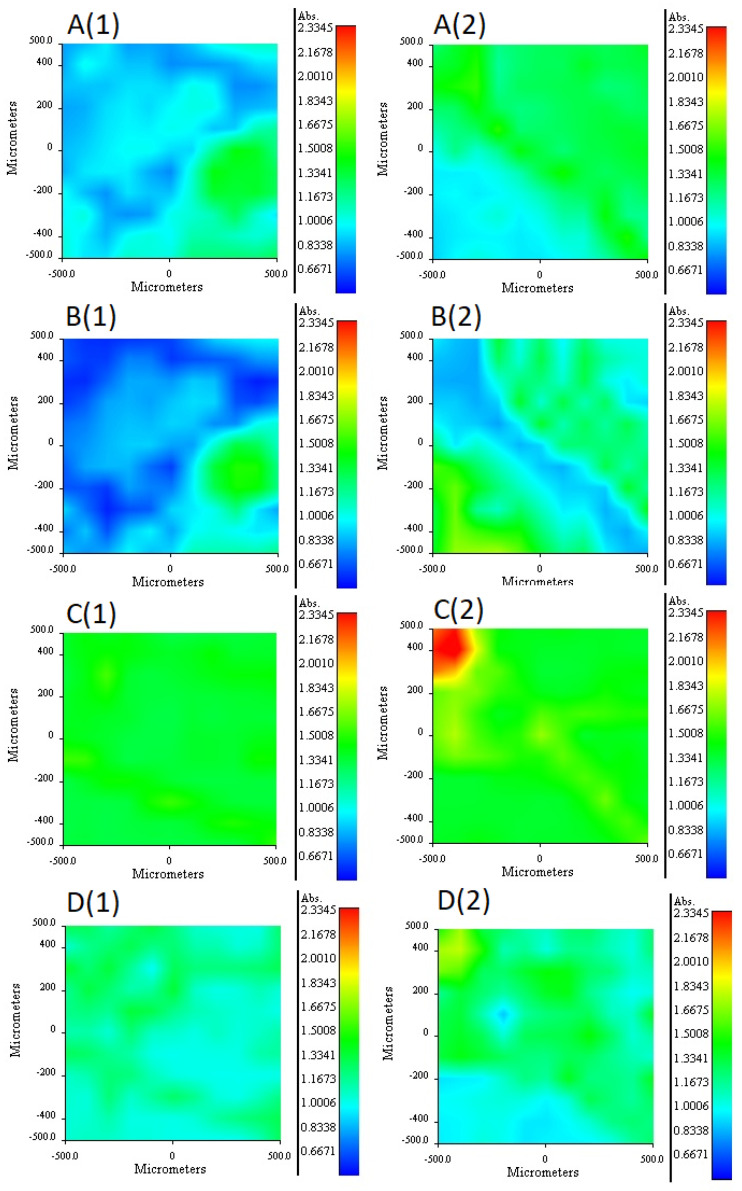
RIMM maps monitoring the distribution of 2NPOE in PIM m2 (**A**), Ionquest^®^ 801 in PIM m2 (**B**), 2NPOE in PIM m4 (**C**), and Ionquest^®^ 801 in PIM m4 (**D**). (1) stands for face 1 and (2) for face 2 of the PIMs (see text).

**Figure 7 membranes-11-00401-f007:**
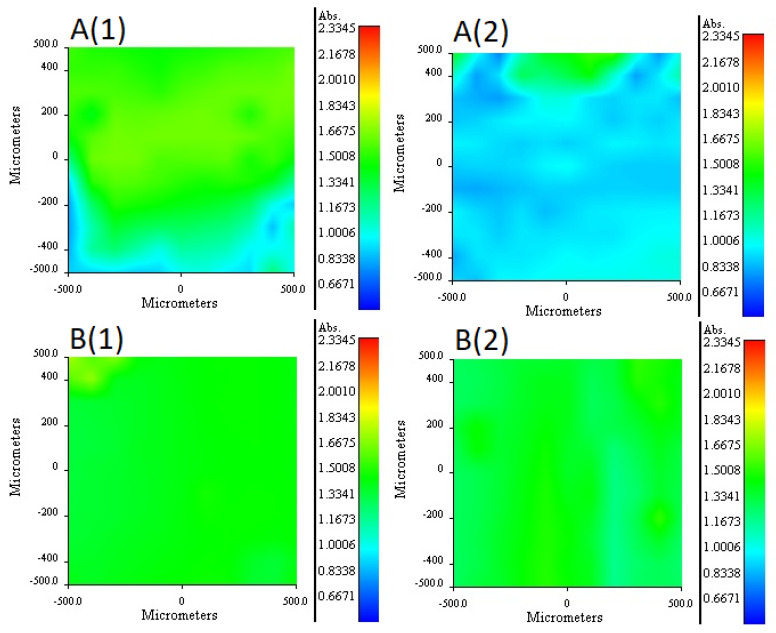
RIMM maps monitoring the distribution of TBEP in PIM m7 (**A**), and TBEP in PIM m9 (**B**). (1) stands for face 1 and (2) for face 2 of the PIMs (see text).

**Figure 8 membranes-11-00401-f008:**
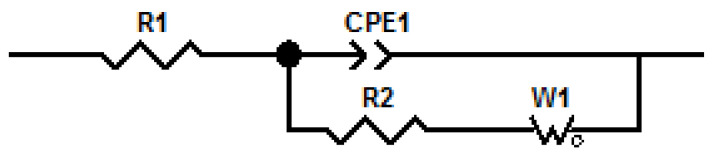
Equivalent circuit to model EIS data. R1 represents the resistance of the solution, R2 the resistance of the membrane, CPE1 is a constant phase element and W1 an open Warburg element.

**Figure 9 membranes-11-00401-f009:**
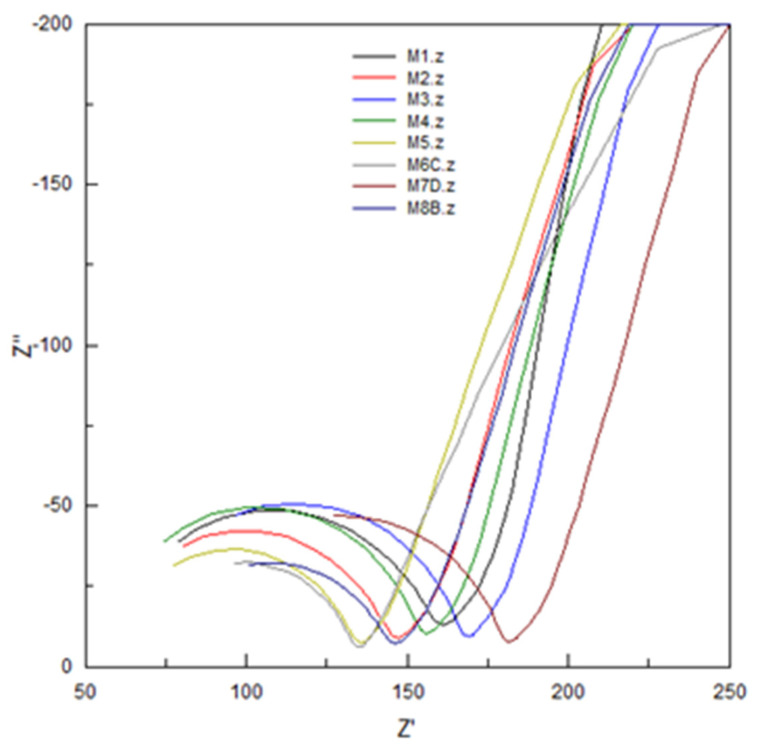
Nyquist plot for EIS characterizations of PIMs, the composition of which is specified in Table 7.

**Figure 10 membranes-11-00401-f010:**
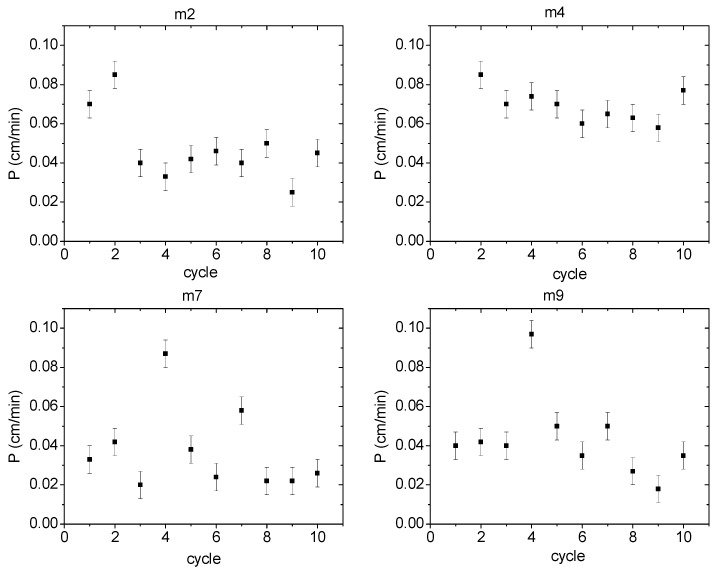
Permeabilities during several operating cycles for PIMs with 2NPOE (m2 (low content) and m4 (high content)) and TBEP (m7 (low content) and m9 (high content)).

**Table 1 membranes-11-00401-t001:** PIM compositions and results of the study of the extraction equilibria by liquid–solid extraction.

PIM	CTA (mg)	Plasticizer (mg)	Ionquest^®^ 801 (mg)	D	Log D
11	30.0	10.5 (2NPOE)	1.0	858.87	2.93
12	30.8	10.3 (2NPOE)	1.4	5046.82	3.70
13	30.2	10.2 (2NPOE)	1.7	11,638.95	4.07
14	30.3	10.2 (2NPOE)	2.1	32,042.25	4.51
15	30.0	10.3 (2NPOE)	2.5	31,892.52	4.50
16	30.5	10.1 (2NPOE)	2.9	31,379.31	4.50
17	30.5	90.3 (2NPOE)	1.0	80.33	1.90
18	30.1	90.1 (2NPOE)	1.4	186.19	2.27
19	30.6	90.1 (2NPOE)	1.9	411.42	2.61
20	30.5	90.0 (2NPOE)	2.3	913.86	2.96
21	30.0	90.1 (2NPOE)	2.6	1341.51	3.13
22	30.1	90.7 (2NPOE)	2.9	1565.91	3.19
23	30.0	10.2 (TBEP)	1.0	1224.26	3.09
24	30.8	9.9 (TBEP)	1.3	2671.96	3.43
25	30.4	10.2 (TBEP)	1.7	6987.13	3.84
26	30.0	10.6 (TBEP)	2.2	31,892.52	4.50
27	30.6	10.6 (TBEP)	2.6	31,164.38	4.49
28	30.6	10.2 (TBEP)	3.1	31,093.39	4.49
29	30.8	90.4 (TBEP)	0.9	205.01	2.31
30	30.8	90.2 (TBEP)	1.5	483.12	2.68
31	29.7	90.3 (TBEP)	1.8	786.28	2.89
32	30.3	90.1 (TBEP)	2.2	1465.38	3.16
33	30.0	90.4 (TBEP)	2.7	2400.94	3.38
34	30.2	90.3 (TBEP)	3.1	2391.23	3.38

**Table 2 membranes-11-00401-t002:** Chemical reaction considered in the numerical evaluation of the extraction equilibria by the LETAGROP–DISTR program.

Chemical Reaction	Equilibrium Constant
${\mathrm{In}}^{3+}+{\mathrm{Cl}}^{-}↔{\mathrm{InCl}}^{2+}$	log β = 2.58
${\mathrm{In}}^{3+}+2{\mathrm{Cl}}^{-}↔{\mathrm{InCl}}^{+}$	log β = 3.84
${\mathrm{In}}^{3+}+3{\mathrm{Cl}}^{-}↔$ InCl_3_	log β = 4.20
$\bar{2\mathrm{HR}}↔\bar{{\left(\mathrm{HR}\right)}_{2}}$	log K_dim_ = 4.09 [27]

**Table 3 membranes-11-00401-t003:** Extraction reactions and their logarithm equilibrium constants for PIMs with 2NPOE (11–16 (low content) and 17–22 (high content)) and TBEP (23–28 (low content) and 29–33 (high content)) obtained after LETAGROP–DISTR analyses.

PIMs	Reaction	log K	U (σ)
11–16	${\mathrm{In}}^{3+}+4\bar{\mathrm{HR}}↔\bar{{\mathrm{InR}}_{3}\mathrm{HR}}+3H^{+}$ ${\mathrm{In}}^{3+}+6\bar{\mathrm{HR}}↔\bar{{\mathrm{InR}}_{3}3\mathrm{HR}}+3H^{+}$	9.16 MAX 9.95	0.18 (0.14)
		10.65 MAX 11.11	
17–22	${\mathrm{In}}^{3+}+5\bar{\mathrm{HR}}↔\bar{{\mathrm{InR}}_{3}2\mathrm{HR}}+3H^{+}$	9.87 ± 0.13	0.05 (0.74)
23–28		8.59 MAX 9.17	0.13 (0.00)
	${\mathrm{In}}^{3+}+3\bar{\mathrm{HR}}↔\bar{{\mathrm{InR}}_{3}}+3H^{+}$ ${\mathrm{In}}^{3+}+6\bar{\mathrm{HR}}↔\bar{{\mathrm{InR}}_{3}3\mathrm{HR}}+3H^{+}$	10.55 MAX 10.87	
29–34		9.47 ± 0.26	0.02 (0.03)
	${\mathrm{In}}^{3+}+4\bar{\mathrm{HR}}↔\bar{{\mathrm{InR}}_{3}\mathrm{HR}}+3H^{+}$ ${\mathrm{In}}^{3+}+6\bar{\mathrm{HR}}↔\bar{{\mathrm{InR}}_{3}3\mathrm{HR}}+3H^{+}$	9.99 MAX 10.60	

**Table 4 membranes-11-00401-t004:** PIM compositions to study In (III) permeability profiles as a function of plasticizer content.

PIM	CTA mg–(%*w*/*w*)–μmol	Plasticizer mg–(% *w*/*w*)–μmol	Ionquest(r) 801 mg–(% *w*/*w*)–μmol
1	30.2–42.4	10.8–15.1–43.0 (2NPOE)	30.3–42.5–98.9
2	30.6–37.7	20.5–25.2–81.6 (2NPOE)	30.1–37.1–98.2
3	30.3–33.2	30.7–33.7–122.2 (2NPOE)	30.2–33.1–98.5
4	30.3–30.0	40.4–40.0–160.7 (2NPOE)	30.4–30.1–99.2
5	30.0–27.0	50.7–45.6–201.7 (2NPOE)	30.5–27.4–99.5
6	30.7–42.6	10.6–14.7–25.0 (TBEP)	30.8–42.7–100.5
7	30.8–37.5	20.5–25.0–48.3 (TBEP)	30.8–37.5–100.5
8	30.5–33.3	30.4–33.2–71.7 (TBEP)	30.7–33.5–100.2
9	30.3–29.9	40.7–40.2–96.0 (TBEP)	30.3–29.9–98.9
10	30.8–27.5	50.6–45.2–119.4 (TBEP)	30.4–27.2–99.2

**Table 5 membranes-11-00401-t005:** Characteristic FTIR vibrations and the corresponding functional groups of the compounds present in the prepared PIMs.

Compound	Band (cm^−1^)	Functional Group
CTA	3600–3200	O–H
CTA	1735	C=O
CTA	1210–1035	C–O–C
CTA	2960–2850	C–H
CTA	1370	C–H
2NPOE	1525	NO_2_
2NPOE	1351	C–N
2NPOE	3480	C–H (aromatic)
2NPOE	2960–2850	–CH_2_–
2NPOE	720	–CH_2_–
2NPOE	1235	R–O–CH_2_
2NPOE	1127	C–O–C
2NPOE	1465	–CH_3_ (octyl)
2NPOE	730–675	C–H
Ionquest^®^ 801 and TBEP	1195	P=O
Ionquest^®^ 801 and TBEP	1038	P–OH (stretch)
Ionquest^®^ 801 and TBEP	1700–2700	P–OH (dimeric form)
TBEP	990	P=O (TBEP)

**Table 6 membranes-11-00401-t006:** PIM compositions for Reflection Infrared Mapping Microscopy characterization.

PIM	CTA (mg)	Plasticizer (mg)	Ionquest^®^ 801 (mg)	Mapped Frequency cm^−1^
39	30.8	–	30.8	801
40	30.7	TBEP (30.6)	–	1136
41	30.6	2NPOE (30.5)	–	1528
m2	30.7	2NPOE (20.2)	30.9	1528 (2NPOE), 976 (Ionquest)
m4	30.3	2NPOE (40.2)	30.2	1528 (2NPOE), 976 (Ionquest)
m7	30.2	TBEP (20.7)	30.2	1136 (TBEP)
m9	30.2	TBEP (40.1)	30.1	1136 (TEBP)

**Table 7 membranes-11-00401-t007:** PIM compositions for EIS characterization.

PIM	CTA (mg)–%*w*/*w*	Plasticizer (mg)–%*w*/*w*	Ionquest^®^ 801 (mg)–%*w*/*w*	Thickness (µm)
M1	(30.2)–80.11	(3.7)–9.81 (2NPOE)	(3.8)–10.08	21.2
M2	(30.8)–45.36	(6.8)–10.01 (2NPOE)	(30.3)–44.62	27.6
M3	(30.4)–33.44	(30.5)–33.55 (2NPOE)	(30.0)–33.00	22.0
M4	(30.4)–45.24	(30.2)–44.94 (2NPOE)	(6.6)–9.82	25.6
M5	(30.1)–79.84	(3.9)–10.34 (TBEP)	(3.7)–9.81	19.2
M6	(30.2)–45.00	(6.6)–9.84 (TBEP)	(30.3)–45.16	25.0
M7	(30.3)–33.30	(30.5)–33.52 (TBEP)	(30.2)–33.19	31.2
M8	(30.6)–45.20	(30.3)–44.76 (TBEP)	(6.8)–10.04	24.0

**Table 8 membranes-11-00401-t008:** Results of the EIS characterization of the PIMs.

PIM	R1 (Ω)	CPE1–T	CPE1–P	R2 (Ω)	W1–R	W1–T	W1–P	ε_r,m_	χ^2^
	PIMs without equilibration with the feed phase								
M1	58.25	3.21 × 10^−9^	1.004	92.27	62.64	0.0001904	0.44279	41.30	0.00023
M2	54.18	4.53 × 10^−9^	0.974	85.03	43.10	0.0002498	0.40866	58.35	0.00015
M3	61.22	2.04 × 10^−9^	1.008	99.92	43.63	0.0001819	0.42318	39.58	0.00019
M4	53.70	2.85 × 10^−9^	1.011	94.02	37.92	0.0001469	0.41267	48.94	0.00016
M5	57.15	4.55 × 10^−9^	0.988	71.93	29.45	0.0001832	0.38955	47.98	8.32 × 10^−5^
M6	51.54	1.84 × 10^−9^	0.852	83.16	1.27	2.252 × 10^−6^	0.3644	54.03	0.00016
M7	76.09	3.20 × 10^−9^	0.958	99.19	39.89	0.0002346	0.41140	56.64	0.00022
M8	65.66	1.16 × 10^−8^	0.899	74.77	31.24	0.0002128	0.39015	57.70	0.00019
	PIMs in contact with a 0.1 mM In (III) solution for 5 min								
M1	50.28	3.07 × 10^−9^	0.992	78.08	31.15	0.0001350	0.41015	48.81	0.00026
M2	60.50	3.81 × 10^−9^	0.985	61.25	41.96	0.0002229	0.42989	81.00	0.00017
M3	65.07	2.55 × 10^−9^	0.984	69.84	28.91	0.0001204	0.40925	56.62	0.00039
M4	57.63	2.45 × 10^−9^	1.025	64.20	35.74	0.0002531	0.43010	71.68	0.00015
M5	54.72	5.11 × 10^−9^	0.994	69.06	38.08	0.0001678	0.36376	49.97	8.32 × 10^−5^
M6	78.57	1.37 × 10^−9^	0.942	46.84	26.20	0.0001156	0.36756	95.94	0.00010
M7	76.15	4.56 × 10^−9^	0.943	88.07	41.85	0.0002402	0.39235	63.68	0.00011
M8	50.66	5.50 × 10^−9^	0.944	82.00	36.62	0.0001884	0.39304	52.61	8.66 × 10^−5^
	PIMs in contact with a 0.1 mM In (III) solution for 180 min								
M1	46.25	3.87 × 10^−9^	0.984	66.77	26.90	0.0002872	0.38189	57.07	0.00013
M2	56.17	5.22 × 10^−9^	0.980	55.89	35.62	0.0002028	0.42254	88.77	9.73 × 10^−5^
M3	60.70	2.57 × 10^−9^	0.992	64.86	30.56	0.0001717	0.39664	60.97	0.00019
M4	56.73	2.59 × 10^−9^	1.020	57.44	33.92	0.0003145	0.38195	80.11	0.00015
M5	52.99	2.24 × 10^−6^	0.988	68.35	45.22	0.0001791	0.40523	50.49	6.61 × 10^−5^
M6	59.02	6.65 × 10^−9^	0.930	75.21	29.90	0.0002680	0.37830	59.75	0.00012
M7	61.58	2.31 × 10^−8^	0.808	92.26	1.01	1.64 × 10^−6^	0.37316	60.79	0.00015
M8	59.77	5.13 × 10^−9^	0.934	75.53	24.37	0.0001011	0.37324	57.12	0.00016

**Table 9 membranes-11-00401-t009:** T_g_ values and permeabilities for PIMs with 2NPOE and TBEP.

PIM	CTA mg–mmol/g	Plasticizer mg–mmol/g	Ionquest^®^ 801 mg–mmol/g	T_g_ (°C)	P × 10^5^ m/s
35	30.5	20.3–0.92 (2NPOE)	35.7–1.35	262.99	
2	30.6	20.5–1.00 (2NPOE)	30.1–1.21		1.36
36	30.3	50.4–1.76 (2NPOE)	32.2–0.93	230.99	
5	30.0	50.7–1.80 (2NPOE)	30.5–0.89		2.11
37	30.3	20.1–0.59 (TBEP)	30.1–1.22	218.93	
7	30.8	20.5–0.59 (TBEP)	30.8–1.22		1.01
38	30.2	42.9–0.98 (TBEP)	32.7–0.97	219.09	
9	30.3	40.7–0.95 (TBEP)	30.3–0.98		0.96

## Data Availability

Not applicable.

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
