# Peer review of "Structural Characterization of the Plasticizers’ Role in Polymer Inclusion Membranes Used for Indium (III) Transport Containing IONQUEST^®^ 801 as Carrier"

_membranes, 2021, doi:10.3390/membranes11060401_

Round 1
Reviewer 1 Report
The submitted article „Structural Characterization of Plasticizers Role in Polymer Inclusion Membranes Used for Indium (III) Transport Containing IONQUEST® 801 as Carrier“ of authors Alejandro Mancilla-Rico, Josefina de Gyves and Eduardo Rodríguez de San Miguel is well written work dealing with effect of action of two (ether-based and phosponic acid ester-based) plasticizers on permeability and stability of prepared PIMs applied for indium (III) separation. The prepared samples of PIMs were well characterized using four instrumental methods (FTIR, RIMM, IES and DSC) and the obtained results were carefully compared and discussed. The selected PIMs samples were compared in In(III) separation process and the role of used plasticizers on PIMs stability is explained. Due to above mentioned reasons this article is appropriate for publishing in Membranes journal.
Only few remarks to the text:
- Please, explain abbreviations/explain in more details chemical structures of CYANEX 272 (line 64) and TOA (line 73).
- Please, explain abbreviation: „SX-based methods“ (line 83).
- Please, correct the phrase: „Rmem y ɛr,m…“ (line 633).
Author Response
Please, explain abbreviations/explain in more details chemical structures of CYANEX 272 (line 64) and TOA (line 73).
R: In the revised version, the chemical names were added in lines 71 (CYANEX 272) and 80 (TOA).
Please, explain abbreviation: „SX-based methods“ (line 83).
R: the acronym SX was introduced in line 40.
Please, correct the phrase: „Rmem y ɛr,m…“ (line 633).
R: In this revised version since the CONCLUSIONS were modified according to the suggestion of Reviewer 2 this sentence no longer appears.
Reviewer 2 Report
In this manuscript, polymer inclusion membranes containing cellulose triacetate as support, Ionquest ® 801 ((2-ethylhexyl acid) -mono (2-ethylhexyl) phosphonic ester) as extractant, and 2NPOE (o-nitrophenyl octyl ether) and TBEP (tri (2-butoxyethyl phosphate)) as plasticizers were characterized using several instrumental techniques. The chemical interactions between the components of the studied PIMs determined crucial aspects of the systems related to membrane formation, physical (thickness), chemical and operational characteristics (permeability and stability). Further, In (III) transport was favored in membranes whose chemical environment was of high polarity and highly depended on a plastic structure that facilitates the permeability of the metal cation. This paper provided some valuable information and the content is very significant in this field. However, I recommended a major revision of the article from its present form before it can be published in membranes. Some specific comments are as follows:
- The abstract and conclusion sections should be a specific and scientific approach.
- In the introduction section, the authors should expound the research significance of the present work.
- The authors should explain the novelty of the present report?
- The authors should provide a schematic representation of the formation mechanism.
- The representation of figures are very poor in quality.
- The pH of the solution normally varies from precursor to precursor. The authors must justify the selection of pH, temperature and time in transport experiments.
- “An amount of 30 mg of CTA was taken as a reference”, why the authors chose this particular quantity.
- What is the key factor (e.g., surface area, chemical composition, morphology) affecting the efficiency?
- In the current state, there are more typographical errors and the language should be improved. Therefore, the authors are advised to recheck the whole manuscript for improving the language and structure carefully.
Author Response
The abstract and conclusion sections should be a specific and scientific approach.
R: As suggested, both sections were rewritten.
In the introduction section, the authors should expound the research significance of the present work.
R: In lines 88-96 of the revised version the significance of the work was addressed:
“This research has been carried out to complement previous works related with In (III) transport in PIMs using CYANEX 272 [14], and D2EHPA (bis(2-ethylhexyl) phosphoric acid) [18] as a comparative study among phosphinic (CYANEX 272), phosphoric (D2EHPA), and phosphonic acid (IONQUEST 801) extractants. Since the influence of the plasticizer in transport is dependent on the degree of plasticity, which is largely dependent on the chemical structure of the plasticizer [19], as well as on its interactions with the other components of the membrane system [14, 18], it is expected that the generated information will be helpful in the understanding of the plasticizer’s role in PIM systems”.
The authors should explain the novelty of the present report?
R: The novelty was remarked in lines 96-99:
“To the best of our knowledge a systematic study of the plasticizer’s influence using different commercial organophosphorus derivatives as carriers in PIMs has not been addressed up to now.”
The authors should provide a schematic representation of the formation mechanism.
R: A reference to a schematic representation was given (reference 24). It was introduced in lines 146-147:
“A description and schematic representation of the evaporation casting method can be found in the literature [24].”
The representation of figures are very poor in quality.
R: Figures 3, 4, and 9 were redrawn.
The pH of the solution normally varies from precursor to precursor. The authors must justify the selection of pH, temperature and time in transport experiments.
R: The selection made was justified in lines 158-159:
“Transport conditions and sampling times were selected according to previously reported similar systems [14, 18].”
“An amount of 30 mg of CTA was taken as a reference”, why the authors chose this particular quantity.
R: This selection was justified in lines 204-206:
“30 mg of CTA was taken as reference, as good mechanical properties and formation characteristics had been previously observed using such amount [14, 18].”
What is the key factor (e.g., surface area, chemical composition, morphology) affecting the efficiency?
R: This observation was addressed in lines 658-667 of the conclusions section:
“In comparison to TBEP, 2NPOE presented then less dispersion and affinity in the PIMs, a plasticizer effect at higher content, higher Rmem and less er,m, and a pronounced drop in the Tg values. However, as In(III) was sorbed by the PIM, these parameters changed and an increase in er,m and a decrease in Rmem were observed, being this effect more pronounced for 2NPOE than for TBEP. In conjunction all the information suggested a better plasticization efficiency of NPOE, which seems to be phase separated, that in the presense of the cation gave rise to a medium of high mobility and polarity, where the structural change promoted by the plasticizer is a key factor in the transport efficiency of the PIM system. A drawback was the decrease in stability because of the minor affinity among the components in 2NPOE–PIMs.”
In the current state, there are more typographical errors and the language should be improved. Therefore, the authors are advised to recheck the whole manuscript for improving the language and structure carefully.
R: The whole manuscript was revised in this aspect.
Finally, some the references were renumbered according to the new order.
Round 2
Reviewer 2 Report
The manuscript can be acceptable in the present form.